

# Evaluation of high-resolution GRAMM/GRAL NO$_x$ simulations over the city of Zurich, Switzerland

Antoine Berchet[1], Katrin Zink[1], Dietmar Oettl[2], Jürg Brunner[3], Lukas Emmenegger[1] and Dominik Brunner[1]

[1]Empa, Swiss Federal Laboratories for Materials Science and Technology, Dübendorf, Switzerland.
[2]Air Quality Control, Government of Styria, Landhausgasse 7, 8010 Graz, Styria, Austria.
[3]Office for Environment and Health protection, City of Zürich, Zürich, Switzerland.

*Correspondence to:* A. Berchet (antoine.berchet@empa.ch)

**Abstract.**

Hourly NO$_x$ concentrations were simulated for the city of Zurich, Switzerland, at 10 m resolution for the years 2013–2014. The simulations were generated with the nested mesoscale meteorology and microscale dispersion model system GRAMM/GRAL (versions v15.12/v14.8) by applying a
catalogue-based approach. This approach was specifically designed to enable long-term city-wide building-resolving simulations with affordable computation costs. It relies on a discrete set of possible weather situations and corresponding steady-state flow and dispersion patterns that are pre-computed and then matched hourly with actual meteorological observations. The modelling system was comprehensively evaluated using eight sites continuously monitoring NO$_x$ concentrations and
65 passive samplers measuring NO$_2$ concentrations on a 2-weekly basis all over the city. The system was demonstrated to fulfil the European Commission standards for air pollution modelling at nearly all sites. The average spatial distribution was very well represented, despite a general tendency to overestimating the observed concentrations, possibly due to a crude representation of traffic-induced turbulence. The temporal variability of concentrations explained by varying emissions and weather
situations was accurately reproduced on different time scales. The seasonal cycle of concentrations, mostly driven by stronger vertical dispersion in summer than in winter, was very well captured in the two year simulation period. Short-term events, such as episodes of particularly high and low concentrations, were detected in most cases by the system, although some unrealistic pollution peaks were occasionally generated, pointing at some limitations of the steady-state approximation. The differ-
ent patterns of the diurnal cycle of concentrations observed in the city were generally well captured as well. The evaluation confirmed the adequacy of the catalogue-based approach in the context of city scale air pollution modelling. The ability to reproduce not only the spatial gradients but also





the hourly temporal variability over multiple years makes the model system particularly suitable for investigating individualized air pollution exposure in the city.

## 1   Introduction

The urban population has grown steadily in the past century and already reached 50% globally and more than 75% in many developed countries. Urban areas with high population density are hot spots of air pollutant emissions, raising concerns regarding increased mortality and morbidity (Cohen et al., 2004; Jerrett et al., 2004; Beelen et al., 2013). Some of the most critical air pollutants in terms of health effects are particulate matter (PM) and $NO_2$, whose levels exceed national and WHO standards in many urban areas (e.g., in Europe; Beelen et al., 2014). In Switzerland, and more particularly in urban centres such as the Zürich area, despite improving trends, the urban population is still exposed to harmful levels of PM smaller than 10 $\mu$m (PM10) and $NO_2$ (Heldstab et al., 2011). Health effects of air pollution are well documented through numerous epidemiological studies (Brunekreef and Holgate, 2002; Beelen et al., 2008; Raaschou-Nielsen et al., 2013), but these studies rely on coarse estimates of the average population exposure as it is very challenging to account for the steep gradients and large temporal variability of air pollutant concentrations in cities (Jerrett et al., 2004; Beelen et al., 2008, 2013). Computing individualized pollution exposure in urban areas requires high-resolution simulations, with at least hourly resolution and spanning long periods of time (years to decades) since health impacts can be triggered by both short-term exceedances of pollution thresholds or long-term continuous exposure to high pollution levels (Van Roosbroeck et al., 2006; Beelen et al., 2008; Lelieveld et al., 2013). Individualized exposure in not only useful for epidemiological studies, but also for air quality plans designed by cities to reduce the direct and indirect social and economic costs of air pollution (e.g., Lelieveld et al., 2013). Current air quality plans are generally lacking a systematic cost-benefit assessment of different mitigation measures due to the lack of affordable model solutions that satisfy the demanding requirements in terms of resolution, temporal coverage, and source-specific information (Miranda et al., 2015).

In the present study, we focus on $NO_x$, an air pollutant with particularly large spatial and temporal gradients due to its short lifetime (e.g., Vardoulakis et al., 2002). Representing the gradients in $NO_x$ concentrations in cities is not yet achievable by standard chemistry-transport models, as they are limited to horizontal resolutions of typically a few kilometres (e.g., Terrenoire et al., 2015). Recent progress in computational fluid dynamics (CFD) models makes it possible to run high-resolution dispersion simulations at the city scale (Li et al., 2006; Kumar et al., 2009, 2011; Di Sabatino et al., 2013). However, the prohibitive computational cost of these simulations prevents their application in the context of long-term urban exposure assessment (Parra et al., 2010). Currently, the most widely used models for urban exposure assessment and regulatory applications are models with a simplified parametrization of pollutant dispersion (e.g. Gaussian plume) such as ADMS (Stocker et al., 2012),



AERMOD (Rood, 2014), SIRANE (Soulhac et al., 2011), IFDM (Lefebvre et al., 2011), or OSPM (Kakosimos et al., 2010). When correctly parametrized and calibrated, these models offer a reliable representation of the average concentration distribution in cities (Soulhac et al., 2011; Briant et al., 2013; Brandt et al., 2013). However, they have difficulties in representing the dispersion in complex building and street canyon configurations and to properly reproduce the temporal (hourly) variability due to varying meteorology (e.g., Soulhac et al., 2012; Ottosen et al., 2015). With the growing availability of urban air pollution observations due to recent advances in (low-cost) sensor technology (Jiao et al., 2016; Gao et al., 2016), land-use regression models (LUR) are increasingly being used for air pollution assessment (Kumar et al., 2015b; Heimann et al., 2015), offering a performance comparable to CFD full physics models (Beelen et al., 2010). Yet, LUR models need a large amount of in-situ observations at strategic locations to represent the full spatial and temporal variability (Duvall et al., 2016; Mueller et al., 2015, 2016; Hasenfratz et al., 2015), and cannot be extended backward in time to satisfy the needs of long-term epidemiological studies.

Considering the respective strengths and limitations of the standard urban air pollution modelling systems, Berchet et al. (2017) proposed a novel method taking advantage of high resolution accurate CFD modelling while keeping computational costs affordable, by using a catalogue-based approach merged with routinely available meteorological observations. They showed that it was computationally feasible to simulate hourly concentration maps over multiple years at building-resolving resolution which successfully capture most of the variability in $NO_x$ concentrations caused by variations in air flow and atmospheric stability. The main purpose of the present study is to provide a comprehensive evaluation of the above-mentioned method for $NO_x$ concentrations in Zürich, Switzerland, for the years 2013–2014. The modelling domain covers the entire urban area of Zurich and includes 8 continuous $NO_x$ monitoring sites as well as 65 $NO_2$ passive samplers. We demonstrate the high quality and robustness of the catalogue-based modelling system for hourly and daily concentrations. Furthermore, we identify sources of errors and uncertainties in the modelling system and propose additional steps to improve the methodology.

In Sect. 2, the modelling chain applied to generate time series of pollution maps is described. In Sect. 3, the set-up for the city of Zurich is presented including the available in-situ observations, the emission inventory, and auxiliary data sets. In Sect. 4, the performance of the model in terms of spatial distribution and temporal variability is evaluated with in-situ $NO_x$ and $NO_2$ measurements.

## 2 Approach and modelling system

### 2.1 Catalogue-based approach

Our approach relies on explicit physical simulations of air flow and pollutant dispersion. Such simulations on a city-wide domain must account for the cascade of scales influencing flow patterns, from the synoptic to the street and building scale. The synoptic scale defines the general meteorological



conditions and the mean direction and strength of the background winds in the city region. Land-use and topography restructure the synoptic weather at the regional scale by generating mesoscale phenomena such as thermally driven land-lake breezes and up- and down-slope circulations, urban heat islands, and channelling and blocking of the flow by the topography. Inside the city, these regional conditions are further modified at the micro-scale by buildings and other obstacles such as vegetation. To properly account for this cascade of scales, our model approach is based on a three-step procedure using the models GRAMM v15.12 (Graz meso-scale model; Almbauer et al., 2000) and GRAL v14.8 (Graz Lagrangian model; Oettl, 2015b), further described in Sect. 2.2: i) mesoscale air flow accounting for topography and land-use effects is computed by GRAMM for a larger domain centered on the city, ii) microscale air flow inside the city, accounting for the effects of buildings on flow and turbulence patterns, is inferred with the GRAL model, forced by GRAMM outputs, and iii) Lagrangian dispersion computations are carried out by the dispersion module of GRAL, constrained by the micro-scale wind fields generated by GRAL. The GRAMM/GRAL system is briefly described in Sect. 2.2.

Simulating the full transient evolution of the atmosphere over a multi-year period is not yet feasible at building resolving resolution (i.e. better than 10 m) for a whole city with current computing resources (e.g., Parra et al., 2010). Therefore, we approximate the full temporal dynamics by a sequence of steady-state solutions selected from a pre-computed catalogue as described in Berchet et al. (2017). This catalogue is a discrete representation of all possible weather situations in terms of atmospheric stability and of large-scale wind speed and direction at the boundaries of the domain. Binning large-scale wind directions and speeds into 36 (10° each) and 7 (from 0.25 to 7 m·s$^{-1}$) categories respectively, with seven possible Pasquill-Gifford classes for atmospheric stability as defined by the U.S. Environment Protection Agency (2000) leads to a catalogue of 1008 physically meaningful reference weather situations. As illustrated in Fig. 1, this catalogue is computed in a three-step procedure which subsequently generates the mesoscale winds computed with GRAMM and the corresponding urban-scale winds and air pollutant concentrations computed with GRAL.

Once the catalogue is available, a sequence of hourly weather situations is built based on in-situ observations of wind speeds and directions in and around the city. For every hour of the simulated period, the weather situation in the catalogue is selected whose associated wind field best matches the in-situ observations. As demonstrated in Berchet et al. (2017), vertical stability and mesoscale flow patterns are intimately linked such that the stability can be sufficiently constrained by matching only the winds at a few selected locations in the model domain. The time series of hourly concentration distributions is then deduced directly from the sequence of weather situations. Transport from remote sources outside of the simulation domain is represented by background concentrations as measured at a rural site near the city, and added to the simulated concentrations. To account for emissions varying independently from the weather, concentration maps in the catalogue are first computed using yearly average emissions and then scaled for each hour according to varying emission activity





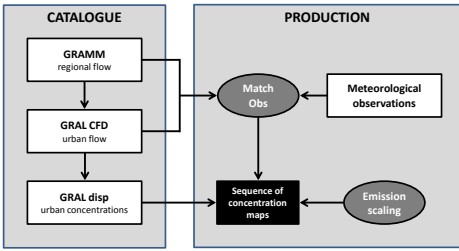

**Figure 1.** Scheme of the multi-step procedure to generate a sequence of hourly steady-state concentration maps from a catalogue of pre-computed wind and concentration fields. The dark grey ellipses are the method steps for generating time series of concentrations from the catalogue. Light grey rectangles denote intermediary products and inputs in the system.

130  (e.g., diurnal cycle of traffic emissions). Since not all types of emissions follow the same temporal profile, emissions are divided into categories (see Sect. 3.3 and Tab. 1), for each of which a catalogue of concentration distributions is computed, and then scaled independently.

Thus, our catalogue-based method can be summarized by the following equation. At a given hour $h$ and point $\mathbf{x}$, the simulated concentration can be written as:

$$c(\mathbf{x},h) = c_{\text{background}}(h) + \sum_{i \in \text{sectors}} \tau_i(h) \times c_i(\eta(h),\mathbf{x}) \qquad (1)$$

where $\tau_i$ is the unitless temporal profile of emissions for each sector $i$, $c_{\text{background}}(h)$ the background concentrations measured at the distant site, and $c_i(\eta(h),\mathbf{x})$ the simulated concentration field for sector $i$ and weather situation $\eta(h)$, obtained with yearly average emissions. The weather situation $\eta(h)$ best matching the meteorological observations at hour $h$ is selected following Eq. 3 of Berchet

140  et al. (2017).

Our modelling approach requires routinely available wind observations in the vicinity of the city of interest, and a background pollution observation site in the rural environment of the city. Berchet et al. (2017) concluded that 5–6 wind observation sites distributed around the city are generally sufficient to represent the variability of weather situations. The emission variability in Eq. 1 is deter-

145  mined from a wealth of information and models, including traffic counts (see Sect. 3) as proxies of emission variability. With such a catalogue-based approach, multi-year hourly physical simulations can be carried out at a cost of only 1–2 months of hourly simulations. The drawback of the approach is that the full transient dynamics is replaced by a sequence of steady-state solutions, but as will be shown in this evaluation, this has only limited impact on the results.



## 2.2 GRAMM/GRAL modelling system

The catalogue-based approach relies on meteorological and on microscale flow and air pollutant dispersion simulations. The mesoscale simulations are carried out by GRAMM (Graz meso-scale model; Oettl, 2015a, 2016) and the microscale simulations by GRAL v14.8 (Graz Lagrangian model; Oettl, 2015b). GRAMM is a non-hydrostatic model solving the conservation equations for mass, enthalpy, momentum, and humidity. It accounts for contrasts in land use and corresponding surface fluxes of heat, momentum and humidity, and it has been specifically designed for operation in steep topography. The large scale weather conditions (wind speed and direction, stability class) of the catalogue are translated into vertical profiles of winds, temperature and pressure, as well as Obukhov length for different stability classes (following the Pasquill-Gifford classification: in the following, from A, very unstable situation, to G, extremely stable) to constrain the initial and boundary conditions of GRAMM simulations.

GRAL is nested into GRAMM and is run here in diagnostic mode at 10 m resolution, which is different from Berchet et al. (2017) where GRAL was run in prognostic mode at 5 m resolution. In diagnostic mode, the flow field around buildings is computed by interpolating GRAMM wind fields on a fine Cartesian grid, and assuming a logarithmic wind profile close to walls. Finally, mass conservation is achieved by applying a Poisson equation to establish a pressure field to correct the velocities. In the prognostic mode, the flow is explicitly computed by forward integration of a set of prognostic equations. We chose here to use the diagnostic mode as the computation costs are lower, which allowed us to simulate a much larger domain covering the complete urban area of Zurich (see Sect. 3.1). We found only minor differences between the simulations with the two modes and resolutions and thus discuss only the results for the diagnostic mode in the following. Lagrangian dispersion simulations are computed with virtual particles released from prescribed emission sources (Oettl and Hausberger, 2006; Oettl, 2014) and transported according to the pre-computed GRAL wind fields. Turbulent diffusion is represented by specific Langevin equations applicable for the full range of wind speeds, in particular for low-wind-speeds (Anfossi et al., 2006).

## 2.3 Evaluation approach

The European Commission expert panel FAIRMODE (Forum for AIR quality MODelling in Europe) has been tasked to define quality objectives and performance criteria for air quality models, following the Directive 2008/50/EC of the European Parliament (EC, 2001). These criteria have been described in Thunis et al. (2012) and Pernigotti et al. (2013). They are base on the following metrics:

- normalized mean bias NMB = $\frac{\bar{S}-\bar{O}}{\bar{O}}$, with $\bar{S}$ and $\bar{O}$ the average simulations and observations, respectively;

- mean fractional bias NFB = $\sum 2\frac{s_i-o_i}{s_i+o_i}$, with $s_i$ and $o_i$ individual simulated and observed values respectively,





– relative percentile error RPE, i.e., the relative error of the 90[th] percentile,

      – FAC2, the fraction of simulations falling into a factor 2 of the observations,

      – MQO, the model quality objective, $\frac{\text{RMSE}}{2U}$, with RMSE denoting the root mean square error and $U$ the average model-observation uncertainty

The performance criteria specify thresholds for the above-mentioned metrics. They are either
defined as an absolute value (50% for RPE and FAC2; 1 for MQO) or are dependent on the model-observation uncertainty. As a pragmatic approach, the uncertainty $U$ was chosen proportional to the observed and simulated values in Thunis et al. (2012). To adapt these definitions to urban simulations in the presence of very high concentration gradients, we add the range of simulated values within a 15-m radius around a given observation site to the uncertainties. Therefore, traffic sites with very
high local gradients have higher uncertainties than urban background sites. Scores are presented for two different simulated time series representing the concentrations at the exact location of the observation sites (reference), and the minimum of all concentrations in a radius of 15 m horizontally and 2 m vertically (minimum), respectively.

Our model is evaluated against these criteria in Sect. 4.1.

## 3   Setup for the city of Zurich

### 3.1   Model domain

The model domain centered on the city of Zurich, Switzerland is illustrated in Fig. 2. The region is characterized by mountain ridges channelling the air flow and separating valleys where the population lives and where air pollution accumulates. In addition, the city of Zurich is located at the
northern extremity of Lake Zurich, an elongated lake that covers an area of $120\,\text{km}^2$, large enough to generate weak land-lake breeze circulations. Mesoscale simulations with GRAMM are carried out in a domain spanning a region of $30 \times 30\,\text{km}^2$, with a horizontal resolution of 100 m. The simulation domain was chosen large enough to allow all topographic features potentially affecting the flow in the city to be represented. The domain extends vertically from the surface to 3000 m above
ground with 22 geometrically spaced layers varying in thickness between 12 m close to the surface and 500 m in the free troposphere.

Dispersion simulations are computed on a smaller domain embracing the whole city of Zurich and most of its outskirts, including some suburban agglomerations and industrial areas. GRAL is run at 10 m horizontal and 2 m vertical resolution on this $17 \times 14\,\text{km}^2$-large domain. Due to the
hilly topography, highways are built through numerous tunnels, creating $NO_x$ emission hotspots at ventilation shafts and tunnel portals, which can optionally be treated in GRAL with a specific algorithm described in Oettl et al. (2002), or simply as point sources at the tunnel gates.





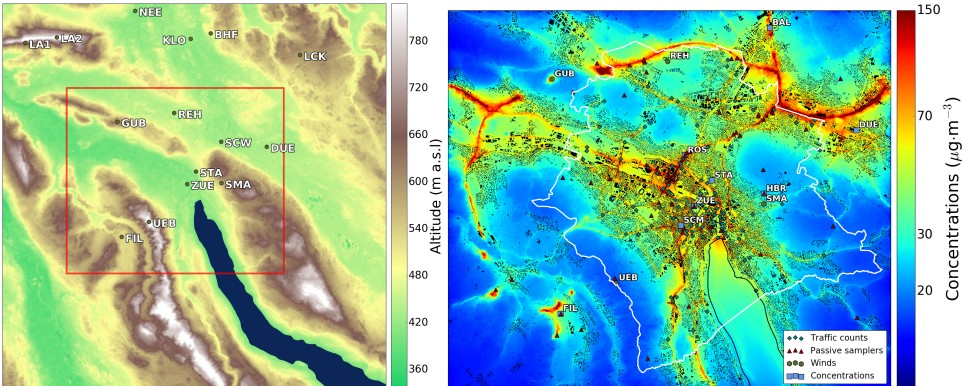

**Figure 2.** Simulation domains for the models GRAMM and GRAL. (left) GRAMM domain and wind measurement sites used in the match-to-observations procedure. The red rectangle is the domain covered by GRAL. (right) GRAL domain with in-situ observation sites overlaid on the 2013–2014 average $NO_x$ concentrations as simulated by the modelling system. The political borders of the city are delimited by the white line.

### 3.2 General model inputs

As mentioned in Sect. 2, air flow computations require information on the topography, land-use
types and buildings. Topographical information was taken from the ASTER GDEM2 data set (Advanced Spaceborne Thermal Emission and Reflection Radiometer – Global Digital Elevation Map Version 2) at a resolution of 30 m and projected to the 100 m GRAMM grid and linearly interpolated to the 10 m GRAL grid. Information on land use (water bodies, forests, etc.) at a resolution of 100 m was taken from the CORINE Land Cover data set (version CLC2006) distributed by the
European Environment Agency. The 44 CORINE land-use classes are translated into typical values for roughness length, heat capacity and thermal conductivity, albedo and soil moisture for GRAMM computations. GRAL uses land-uses classes in terms of roughness length to account for surface drag caused by different types of vegetation, whereas the drag imposed by buildings is represented explicitly. The CORINE data set is projected similarly to ASTER to the GRAMM and GRAL grids.
Three-dimensional building information inside the city of Zurich was deduced from a vectorial building inventory provided by the municipality of Zurich. Buildings outside of the city are taken from the nation-wide vectorial data base swissBUILDINGS3D v2.0, provided by the Swiss Federal Office of Topography, Swisstopo. Vectorial building shapes were projected to the 10 m GRAL grid, i.e, buildings are represented by individual blocks of 10 m × 10 m horizontal size.

### 235  3.3 Emission data

Emission data are deduced from two very detailed inventories produced by the municipal (Umwelt-
und Gesundheitsschutz, UGZ) and cantonal (Amt für Abfall, Wasser, Energie und Luft, AWEL)





| Emission type | Simulated category | Inventory category | Resolution | Total emission t.y$^{-1}$ / % |
|---|---|---|---|---|
| Light duty traffic | UGZ light | Cars | 7500 lines | 495 / 18.0 |
| | | Motorbikes | 7500 lines | 5.7 / 2.7 |
| | AWEL light | Cars | 15500 lines | 271.6 / 9.9 |
| | | Motorbikes | 15500 lines | 4.5 / 0.2 |
| | AWEL area traffic | Non-road traffic | 100 m grid | 233.4 / 8.5 |
| Heavy duty traffic | UGZ heavy | Heavy duty traffic | 7500 lines | 321.7 / 11.7 |
| | UGZ bus | Buses | 7500 lines | 58.9 /2.1 |
| | AWEL heavy | Heavy goods vehicles | 15500 lines | 154.9 / 5.6 |
| | | Local delivery | 15500 lines | 77.8 / 2.8 |
| | AWEL bus | Local buses | 15500 lines | 27.0 / 1.0 |
| | | Long-distance buses | 15500 lines | 13.8 / 0.5 |
| Heating systems | UGZ heating | Oil boilers | 10500 points | 251 / 9.1 |
| | | Gas boilers | 13200 points | 138 / 5.0 |
| | UGZ boilers | Hot-water generators | 620 points | 2.7 / 0.1 |
| | UGZ wood | Wood-burning systems | 900 points | 18.7 / 0.7 |
| | AWEL heating | Oil-gas systems | 100 m grid | 145.0 / 5.3 |
| | AWEL wood | Wood-burning systems | 100 m grid | 26.6 / 1.0 |
| Industry | UGZ industry | Medium-size industries | 270 points | 29.5 / 1.1 |
| | | Waste-burning and heat plants | 56 points | 175 / 6.3 |
| | UGZ off-road | Construction machines | vector areas | 75.5 / 2.8 |
| | AWEL industry | Medium and large industries | 100 m grid | 86.3 / 3.1 |
| | | Industrial vehicles | 100 m grid | 100.8 / 3.7 |
| | | Smaller industrial emissions | 100 m grid | 4.8 / 0.2 |
| Ships | UGZ ships | lake-cruise boats | cruise lines | 20.9 / 0.8 |
| | UGZ private boats | privately-owned boats | lake area | 5.6 / 0.2 |
| Total | | | | 2744.4 / 100 |

**Table 1.** Description of NO$_x$ emission inventories in Zurich as used in GRAL simulations. Emissions inside the limits of the city of Zurich are provided by UGZ, emissions for the rest of the domain by AWEL.





environment authorities. Inventory information are detailed in Tab. 1. Both inventories are spatially explicit bottom-up inventories based on activity data and emission factors as detailed in e.g., FOEN (2010) and Heldstab et al. (2016). The two inventories have been designed for the year 2010 and are highly consistent in terms of total emissions over the domain of Zurich. $NO_x$ emissions at the national scale are reported to have decreased by 5–10% depending on the emission category between 2010 and the period of our simulations. We assume that city emissions follow the national trend and apply corresponding correction factors separately to the individual emission categories.

The UGZ inventory details emissions from thousands of individual sources as line, point or area sources divided into 60 emission categories (cars, motorbikes, gas heating systems, wood heating systems, etc.) within the city limits. Although accounted for in the UGZ inventory, some of these emission categories have a very marginal contribution to the $NO_x$ emissions (e.g., forestry and agriculture machines, smokers, animals), or are very punctual (e.g., fireworks), and have therefore been ignored in our simulations. These neglected emissions account for less than 1% of total $NO_x$ emissions. Individual heating systems are registered by type and size and by the exact location and elevation of the chimney and treated as 26000 individual point sources. Emissions from cars, motorcycles, lorries and buses are represented as line sources segmented into 5 to 50 m-long segments and are based on a comprehensive traffic-emission model and manual traffic-counting campaigns; tunnel portal are modelled as point sources integrating traffic emissions inside the tunnel. The AWEL inventory is less detailed but covers areas that are outside of the city but still inside the GRAL domain. It describes 20 different emission categories as line or area sources. Main roads are described as line sources, while other sources are represented as area sources with a 100 m resolution grid. AWEL emissions are disaggregated to the GRAL grid using the building mask to attribute heating and industrial emissions to building roofs and other emissions to the space between buildings.

As GRAL can account for the rise of hot plumes in ambient air by applying a slightly modified version of the plume-rise model described in Hurley et al. (2005), an initial exhaust temperature and speed is prescribed for point emissions. Such values are available only for the biggest emitters such as waste incineration plants. For all other heating systems, a standard temperature and exhaust speed of $70°$ C and $0.8$ m·s$^{-1}$ with a stack diameter of $0.5$ m was prescribed. To account for the turbulence induced by the traffic at least to first order, emissions from car and heavy duty traffic are initially mixed within a volume defined by the width of the street (uniformly set at 7 m) and a height of 3 m above street level.

To limit the computational demand, we merged the original categories into a total of 25 group-categories by adding up emissions with a similar temporal profile. For instance, we expect motorbike emissions to vary similarly to car emissions. Emission variability in Eq. 1 is determined based on both pre-defined profiles and measured proxies. For all computed emission categories, we apply typical diurnal, weekly and seasonal cycles as used in the TNO-MACC emission inventory for Europe (Kuenen et al., 2011), with the exception of light duty traffic and heating emissions. 85 traffic





counts are operated by the municipality in the city of Zurich. We use the hourly ratios of the total
number of vehicles (summed over all sites) to the annual average hourly total. Heating emissions
follow a diurnal cycle as prescribed in the TNO-MACC emission inventory, but the seasonal cycle
of such emissions is determined using so-called "heating-degree days" accounting for the outdoor
temperatures as measured at different locations in the city. Heating degree days are computed at the
daily scale using Eq. 2:

$$
\begin{aligned}
\mathrm{HDD}(t) &= \mathrm{T_{ref}} - \mathrm{T}(t) \quad \text{if } \mathrm{T}(t) < \mathrm{T_{min}} \\
&= 0 \qquad\qquad \text{else}
\end{aligned}
\tag{2}
$$

with $\mathrm{T_{ref}} = 20°\mathrm{C}$ and $\mathrm{T_{min}} = 16°\mathrm{C}$ and $\mathrm{T}(t)$ the daily average outdoor temperature in the city at time
$t$.

Heating emissions are scaled proportionally to the heating degree days parameter. As the total
number of heating degree days varies from one year to another, depending on the meteorology, the
scaling factor for heating emissions is chosen to keep consistent heating degree days and emissions
for the year 2010 for which the inventory was designed.

### 3.4  Meteorological and air pollution observations

Meteorological observation sites used for the match-to-observations procedure are shown on the
GRAMM domain of Fig. 2. Wherever possible, weather observations are compared with GRAL
wind simulations as they are able to represent the influence of nearby obstacles on the air flow.
Outside of the GRAL domain, GRAMM mesoscale simulations are used, which limits the selection
to standard weather observation sites in open terrain following WMO recommendations as operated
by the Swiss Federal Office of Meteorology and Climatology, MeteoSwiss. The remaining weather
observations are obtained from air pollution observation sites, maintained by the Swiss national
air pollution monitoring network, NABEL, the regional monitoring network for East-Switzerland,
OSTLUFT, and the city environment authority, UGZ. At all MeteoSwiss sites but UEB and LA2,
wind speeds and directions are measured on top of a 10-m tall meteorological mast. At UEB and
LA2, wind measurements are carried out on top of a 189 m- and 32 m high telecommunication tower,
respectively. Weather observation sites within the city are located above building roofs or in street
canyons.

Eight $NO_x$ concentration measurement sites used for evaluating the model are operated within
the GRAL domain. GRAL concentrations in the corresponding cells are used for the assessment
of the model performance. The rural site TAE, operated by NABEL 25 km away from the GRAL
domain boundaries, is taken as regional background site. The site ZUE is located in the centre of
Zurich in a courtyard, distant from emission hot spots. The sites BAL, ROS, STA and SCM are
located next to busy streets. DUE is operated in the outskirts of the city in a mixed industrial and
residential area. HBR is located on the Zurichberg mountain, 200 m above the city centre level, at



the limit of the built-up area next to a forest. At all sites, the air inlet is located at 3–4 m above
ground level. $NO_x$ concentrations are measured with standard $NO_x$ monitors: Horiba APNA 360 and
Horiba APNA 370. The Horiba APNA instruments use molybdenum converters to convert $NO_2$ into
NO before measuring NO by chemiluminescence. These instruments are therefore also sensitive to
other reactive nitrogen compounds, which may lead to some bias in the $NO_2$ measurements during
periods when $NO_2$ concentrations are low but the concentrations of ozone and other photo-oxidants
high (Steinbacher et al., 2007). Such bias are expected to impact significantly the background site
TAE as discussed in Sect. 4.2. The instruments are automatically calibrated every 25 h and manually
every 2 weeks at NABEL sites, and every 10 days at UGZ and OSTLUFT sites. The calibration is
carried out through dynamic dilution of a certified NO mixture (Carbagas).

All the sites mentioned above provide continuous hourly measurements for the simulated period
from January 2013 to December 2014. The monitoring network is complemented by a network of
65 passive $NO_2$-samplers maintained by UGZ and OSTLUFT. These samplers are collected every
2 weeks and analysed in the laboratory shortly after collection. For making the biweekly samples
comparable to the continuous observations, a few passive samplers are placed next to continuous
sites. $NO_2$ observations from all passive samplers are then corrected with a linear correction function
by comparing continuous and passive measurements at these sites.

Temperature data used to scale emissions from heating systems is gathered at the same locations
as the wind data. The average outdoor temperature for the entire city is calculated as the mean of
available observations. Traffic counts are located all around the city. They count vehicles indifferent
of their type on a 15-minute-basis direction-wise for all lanes of selected streets. We use hourly totals
in the city to scale traffic emissions uniformly.

## 4   Results

After generating the catalogue of wind and concentration fields, hourly time series of concentration
maps have been generated for the years 2013 and 2014. In the following, these model outputs are
evaluated against observations and an analysis of uncertainties of the model system is presented.

### 4.1   General model performance

Our model is evaluated at all sites against the FAIRMODE performance criteria as defined in Sect. 2.3.
All FAIRMODE scores are reported in Tab. 2. In the reference simulation, most sites fulfil the crite-
ria. No more than $\sim 15\%$ of all computed scores are beyond the FAIRMODE performance criteria.
The few exceptions are discussed below.

At the site ROS, the model largely overestimates the concentrations, resulting in poor scores for
NMB, FAC2, RPE and MQO. This can be explained by the very steep gradients in the vicinity of
the site. ROS is located on a small parking lot, adjacent to the busiest traffic corridor in the city.



| Temporal scale | Simulation ID | Site ID | Observed mean ($\mu g \cdot m^{-3}$) | 15-m radius range ($\mu g \cdot m^{-3}$) | NMB (%) | | MFB (%) | | RPE (%) | | FAC2 (%) | | r | | MQO |
|---|---|---|---|---|---|---|---|---|---|---|---|---|---|---|---|
| Hourly | Ref | HBR | 23.1 | 7.1 | 46 | 29 | 46 | 30 | 50 | 27 | 50 | 78 | 0.66 | 0.78 | 0.81 |
| | | FIL | 38.1 | 6.6 | 37 | 36 | 37 | 29 | 50 | 36 | 50 | 64 | 0.78 | **0.61** | **1.37** |
| | | DUE | 45.1 | 10.3 | 43 | 40 | 43 | 43 | 50 | 33 | 50 | 61 | 0.78 | **0.62** | **1.31** |
| | | ZUE | 46.6 | 12.9 | 42 | 26 | 42 | 28 | 50 | 23 | 50 | 76 | 0.68 | 0.73 | 0.93 |
| | | STA | 63.9 | 30.3 | 53 | 29 | 53 | 33 | 50 | 23 | 50 | 73 | 0.46 | 0.75 | 0.68 |
| | | BAL | 93.5 | 70.8 | 77 | 25 | 77 | 9 | 50 | 35 | 50 | 66 | 0.0 | 0.53 | 0.68 |
| | | SCM | 95.9 | 58.7 | 65 | 52 | 65 | 36 | 50 | **62** | 50 | 68 | 0.27 | 0.66 | 0.80 |
| | | ROS | 119.8 | 331.2 | 69 | **135** | 69 | 67 | 50 | **165** | 50 | **43** | 0.23 | 0.62 | **1.07** |
| | | ROS2 | 119.8 | 322.7 | 69 | **79** | 69 | 45 | 50 | **102** | 50 | 56 | 0.0 | 0.60 | 0.79 |
| | | All | 65.3 | 57.3 | 106 | 51 | 106 | 33 | 50 | **52** | 50 | 67 | 0.0 | 0.67 | 0.66 |
| | Min | HBR | 23.1 | 7.1 | 46 | 8 | 46 | 14 | 50 | 8 | 50 | 85 | 0.66 | 0.79 | 0.78 |
| | | FIL | 38.1 | 6.6 | 37 | 18 | 37 | 18 | 50 | 17 | 50 | 68 | **0.78** | 0.64 | **1.25** |
| | | DUE | 45.1 | 10.3 | 43 | 22 | 43 | 31 | 50 | 15 | 50 | 66 | **0.78** | 0.64 | **1.23** |
| | | ZUE | 46.6 | 12.9 | 42 | -8 | 42 | 5 | 50 | -15 | 50 | 83 | 0.68 | 0.76 | 0.92 |
| | | STA | 63.9 | 30.3 | 53 | -11 | 53 | 1 | 50 | -17 | 50 | 83 | 0.46 | 0.79 | 0.63 |
| | | BAL | 93.5 | 70.8 | 77 | -6 | 77 | -13 | 50 | 3 | 50 | 62 | 0.0 | 0.58 | 0.53 |
| | | SCM | 95.9 | 58.7 | 65 | -4 | 65 | -1 | 50 | -2 | 50 | 81 | 0.27 | 0.72 | 0.52 |
| | | ROS | 119.8 | 331.2 | 69 | 17 | 69 | 9 | 50 | 31 | 50 | 70 | 0.23 | 0.58 | 0.52 |
| | | ROS2 | 119.8 | 322.7 | 69 | 18 | 69 | 10 | 50 | 32 | 50 | 70 | 0.0 | 0.58 | 0.53 |
| | | All | 65.3 | 57.3 | 106 | 1 | 106 | 7 | 50 | -1 | 50 | 76 | 0.0 | 0.71 | 0.38 |
| Daily | Ref | HBR | 23.1 | 7.0 | 39 | 28 | 39 | 32 | 50 | 19 | 50 | 92 | 0.60 | 0.93 | 0.41 |
| | | FIL | 36.0 | 6.1 | 31 | **33** | 31 | 31 | 50 | 37 | 50 | 80 | 0.73 | 0.83 | 0.79 |
| | | DUE | 44.6 | 10.2 | 36 | **40** | 36 | 43 | 50 | 25 | 50 | 72 | 0.70 | 0.85 | 0.71 |
| | | ZUE | 46.7 | 13.0 | 37 | 26 | 37 | 30 | 50 | 21 | 50 | 90 | 0.60 | 0.87 | 0.58 |
| | | STA | 64.0 | 30.4 | 46 | 29 | 46 | 32 | 50 | 20 | 50 | 87 | 0.20 | 0.87 | 0.41 |
| | | BAL | 93.3 | 69.8 | 52 | 24 | 52 | 13 | 50 | 39 | 50 | 86 | 0.0 | 0.70 | 0.52 |
| | | SCM | 96.0 | 58.9 | 56 | 52 | 56 | 39 | 50 | **60** | 50 | 78 | 0.0 | 0.81 | 0.47 |
| | | ROS | 119.8 | 331.7 | 48 | **135** | 48 | **78** | 50 | **151** | 50 | **31** | 0.07 | 0.75 | 0.78 |
| | | All | 64.6 | 56.0 | 85 | 51 | 85 | 36 | 50 | **63** | 50 | 79 | 0.0 | 0.80 | 0.54 |
| | Min | HBR | 23.1 | 7.0 | 39 | 7 | 39 | 14 | 50 | 1 | 50 | 98 | 0.60 | 0.93 | 0.44 |
| | | FIL | 36.0 | 6.1 | 31 | 15 | 31 | 17 | 50 | 18 | 50 | 88 | 0.73 | 0.84 | 0.76 |
| | | DUE | 44.6 | 10.2 | 36 | 22 | 36 | 31 | 50 | 8 | 50 | 84 | 0.70 | 0.86 | 0.72 |
| | | ZUE | 46.7 | 13.0 | 37 | -7 | 37 | 2 | 50 | -15 | 50 | 98 | 0.60 | 0.88 | 0.69 |
| | | STA | 64.0 | 30.4 | 46 | -11 | 46 | -4 | 50 | -18 | 50 | 98 | 0.2 | 0.89 | 0.45 |
| | | BAL | 93.3 | 69.8 | 52 | -7 | 52 | -13 | 50 | 5 | 50 | 88 | 0.0 | 0.73 | 0.39 |
| | | SCM | 96.0 | 58.9 | 56 | -4 | 56 | -3 | 50 | -2 | 50 | 97 | 0.0 | 0.84 | 0.33 |
| | | ROS | 119.8 | 331.7 | 48 | 17 | 48 | 17 | 50 | 14 | 50 | 97 | 0.07 | 0.74 | 0.34 |
| | | All | 64.6 | 56.0 | 85 | 1 | 85 | 6 | 50 | 2 | 50 | 94 | 0.0 | 0.86 | 0.26 |

**Table 2.** Model performance for $NO_x$ concentrations at all observation sites at the hourly and daily scales. Ref = standard simulation at the exact location of observation sites; Min = simulated minimum in a radius of 15 m horizontally and 2 m vertically. The score metrics are defined in Sect. 2.3. The left column for each score represents the FAIRMODE performance objective (defined in Sect. 2.3) following Thunis et al. (2012) dependent on $U$; the right column is the computed score. Values not fulfilling the FAIRMODE objective are reported in bold. Sites are sorted by increasing observed mean.





Simulated concentrations vary spatially by more than $300\,\mu\mathrm{g\cdot m^{-3}}$ within a radius of $15\,\mathrm{m}$ around
the exact location of the site, compared to the observed average of $120\,\mu\mathrm{g\cdot m^{-3}}$. Small errors in
the location of the emissions relative to the site or in the computation of the flow fields have a
critical impact on such a site. At the site SCM, the relative percentile error exceeds the requirement
threshold due to overestimated concentration peaks. The site SCM is a traffic site similar to ROS,
with likely similar reasons for overestimation. However, an overestimation at SCM is only evident
for the highest concentrations whereas at ROS the concentrations are generally too high. The sites
FIL and DUE do not fulfil all FAIRMODE performance objectives as well, but in their case due to
insufficient correlation between observations and simulations at the hourly scale. The site DUE is
located in an industrial area, next to a busy highway, while FIL is located in a rural environment
but next to a motorway intersection and the portals of the three tunnels of the motorway bypass in
Zurich West. Incorrectly prescribed variability in related traffic and industrial emissions may explain
the insufficient scores. Discrepancies between observations and simulations are further discussed in
the following sections to identify error sources and propose possible model improvements.

Though fulfilling the quality objectives in most cases, the modelling system seems to generally
overestimate concentrations at all observation locations. The overestimation might be attributed to
incorrect emission magnitude in the inventories. However, when considering the minimum values
within a $15\,\mathrm{m}$ horizontal and $2\,\mathrm{m}$ vertical distance, all performance scores at all sites are signifi-
cantly improved. The biases almost vanish and the synthesis MQO index is smaller than 1 at almost
all sites at the hourly and daily scale, and even below 0.5 on average. A MQO below 0.5 was consid-
ered by Thunis et al. (2012) as a reference objective since observation errors start dominating model
error below 0.5. The general improvement of the model performance when taking the minimum in
a certain distance rather than values at the exact location of the observations suggests that the dis-
persion of pollutants is generally underestimated, especially in the vicinity of emission hot spots.
In fact, GRAL is known for overestimating pollutant concentrations near building façades (Oettl,
2015b). In addition, traffic-induced turbulence is accounted for only by spreading traffic emissions
over the lowest $3\,\mathrm{m}$ above ground (see Sect. 3.3). The better performance of the model in the mini-
mum simulation at traffic sites suggests that additional efforts should be made to better parametrize
the traffic-induced turbulence.

The temporal correlations between simulations and observations do not change drastically be-
tween the two approaches, but also for this metric there is an improvement when using the minimum
values in most cases. The correlations between observations and simulations are in the range 0.53–
0.79 at the hourly scale and 0.71–0.94 at the daily scale. At most sites, these correlations are at
least 0.5 higher than the correlations between observations and temporally varying emissions, which
demonstrates that meteorological variability is a key factor driving the variability in concentrations
and that this variability is very well captured by the catalogue-based modelling approach. Excep-
tions are the sites ROS, SCM and BAL for which the correlations are only 0.05 to 0.2 better than



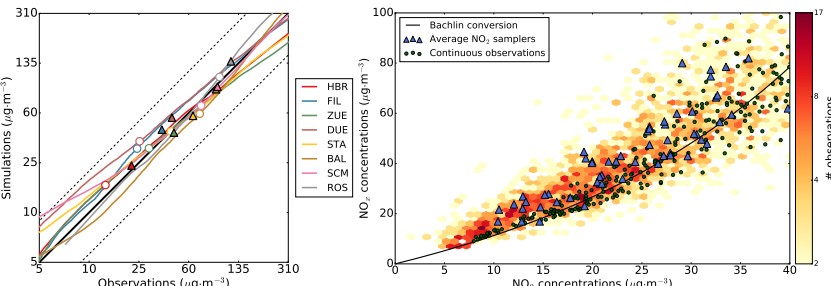

**Figure 3.** Comparison of observed and simulated $NO_x$ and $NO_2$ concentrations. (left) Quantile-quantile plot of hourly $NO_x$ concentrations at continuous sites. The solid and dashed black lines are the 1:1 line and the 1:2 and 2:1 lines, respectively. Filled triangles are the 2013–2014 mean and empty circles the median concentrations. Note the logarithmic scale of the plot. (right) Concentrations at passive samplers. Yellow to red coloured background show the comparison between 2-weekly observed $NO_2$ (x-axis) and corresponding simulated $NO_x$ (y-axis). Triangles are the mean values per station averaged over the entire period 2013–2014. For comparison, green dots are the two-weekly averages of $NO_2$ and $NO_x$ as observed at continuous sites. The black line is a "Bachlin"-type parametrization of the ratio between $NO_x$ to $NO_2$ (Düring et al., 2011).

the emission-observation correlations. At these traffic sites the variability appears to be dominated by traffic intensity rather than by meteorology.

In the following, we compare observations to the "minimum" simulations due to their significantly better performance, and will further discuss the implications of this choice in Sect. 5.

### 4.2   Evaluation of the spatial distribution

The average distribution of simulated $NO_x$ concentrations is shown in Fig. 2. Apart from hills and forest areas, where the concentrations are close to the background, $NO_x$ concentrations are dominated by local emissions in most built-up areas. Large gradients exist between traffic corridors with concentrations higher than $100\,\mu g \cdot m^{-3}$ and backyards and smaller streets.

To evaluate the quality of this average distribution, we use eight continuous monitoring sites and
65 $NO_2$ passive samplers distributed rather uniformly over the city and covering the full range of pollution levels. Figure 3 and Tab. 2 compare average observations with simulations. As shown by the quantile-quantile plot and the biases, there is no specific dependency of the mismatches on the concentrations. The fractional bias remains roughly the same (well below 50%) at all sites over the whole range of observed concentrations. As for the biases, it is considered that an air pollution model
is performing well when the NMB is below 50% (e.g., Kumar et al., 2006). The relative bias only seems to increase at all sites at the lower and upper end of the concentration range, suggesting higher uncertainties for very high and very low concentrations.



At passive sampler sites the comparison is complicated by the fact that the modelling system simulates $NO_x$ whereas passive samplers measure $NO_2$. The ratio between $NO_2$ and $NO_x$ is of-

ten parametrized by a non-linear "Bachlin" function depending solely on the concentration of $NO_x$ (e.g., Düring et al., 2011). The function accounts for the fact that the ratio tends to increase with increasing distance from the source and hence with decreasing $NO_x$ concentration. The ratios between biweekly averaged $NO_2$ and $NO_x$ concentrations as measured at the continuous sites (green dots in Fig. 3) indeed closely follow a Bachlin curve, though with increasing spread at high $NO_x$

concentrations. The ratios between the two-year averages of $NO_2$ measured at the passive samplers and the corresponding simulated $NO_x$ values follow this curve as well, although the simulated $NO_x$ values tend to be somewhat too high. For the individual 2-weekly averages (coloured background in right panel of Fig. 3), this overestimation is particularly evident for the low concentrations. These low concentrations mainly occur during the summer season when the lifetime of $NO_x$ is shortest.

The overestimation of low concentrations could therefore be a result of treating $NO_x$ as a passive tracer in the model not accounting for photochemical depletion. Furthermore, Steinbacher et al. (2007) demonstrated that $NO_x$ concentrations at TAE, which are added as a background to our simulations, are generally overestimated by 3–4 $\mu g \cdot m^{-3}$ due to interferences of other reactive nitrogen compounds like PAN and $HNO_3$ when using a molybdenum converter.

Apart from a possible overestimation of very low concentrations, our modelling system is able to reproduce the large spatial variations in average concentration levels with high confidence.

### 4.3 Evaluation of the temporal variability

Good scores for the average spatial distribution of air pollutant concentrations have already been demonstrated for other modelling systems (e.g., Soulhac et al., 2011; Di Sabatino et al., 2007).

However, accurately reproducing not only average concentrations but also the temporal evolution from hourly to seasonal time scales is a much more challenging objective that has received little attention so far. This section therefore focusses on evaluating the simulated temporal variability.

#### 4.3.1 Example period

In Fig. 4, observations are compared to simulations for a selected period of time, October 2013. The

period has been selected as the only time, when all sites were in operation, but also because the concentrations represented the average patterns of concentration variability rather than some specific pollution events. The simulations are divided into their main contributors (background, light and heavy duty traffic, heating systems and the rest). Sites are sorted following the average observed $NO_x$ concentrations, from the highest at ROS to the lowest at HBR. Simulations at all sites are gen-

erally in very good agreement with observations: the diurnal cycle is mostly well reproduced and the different concentration patterns during weekends (shaded periods in Fig. 4) are well detected. Whereas background concentrations are the same at all sites and never exceed 50 $\mu g \cdot m^{-3}$, the mag-



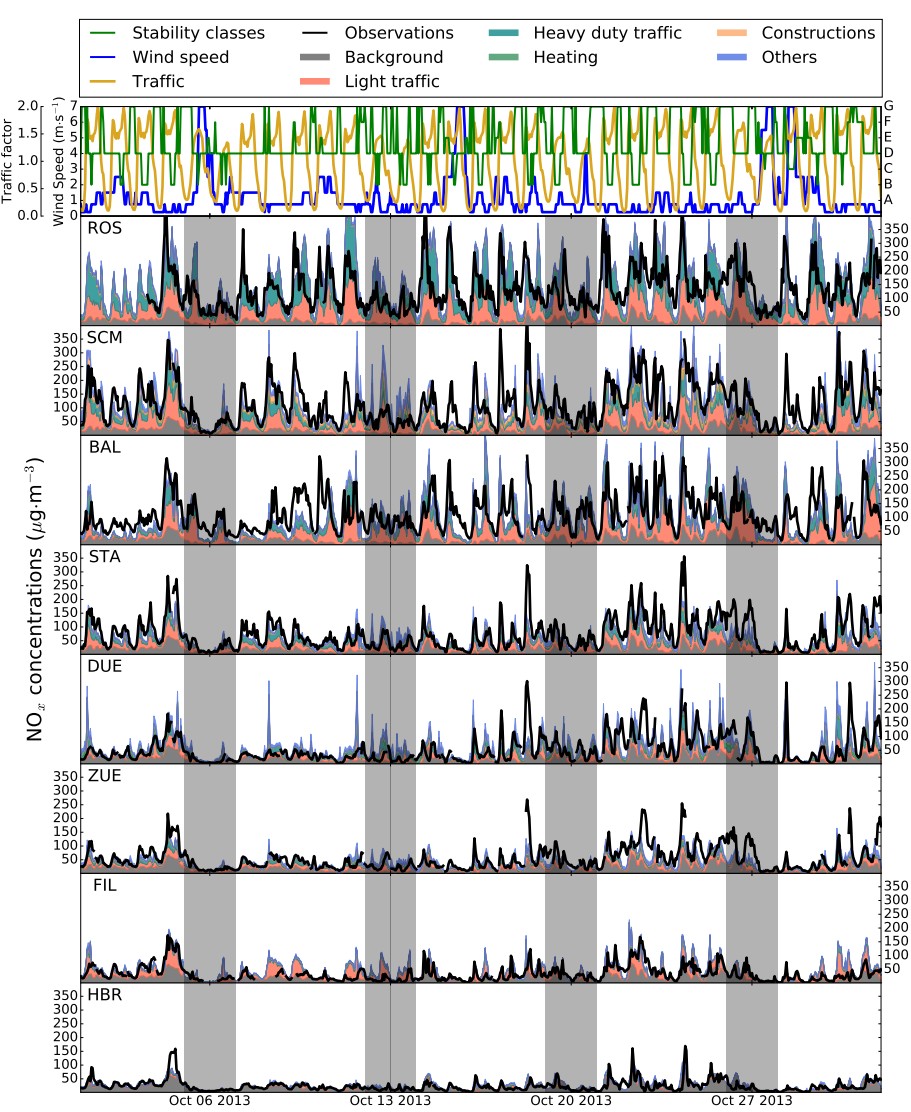

**Figure 4.** Hourly time series of observed and simulated concentrations at all sites for the month October 2013. Simulations are separated by contributions from the main emission categories (light traffic, heavy duty traffic, heating and the rest). Shaded periods represent week-ends. (top) Stability classes and wind categories of the hourly selected weather situations.





nitude of local contributions varies significantly from one site to the other, very consistently with observations. FIL and HBR show almost no local contribution whereas street sites such as ROS and

SCM are largely dominated by local traffic emissions. However, at the three most polluted sites, ROS, SCM and STA, the simulations deviate significantly from the observations for some periods. As discussed earlier this is likely related to the large local gradients at these sites of $30\,\mu\mathrm{g\cdot m^{-3}}$, $59\,\mu\mathrm{g\cdot m^{-3}}$ and $330\,\mu\mathrm{g\cdot m^{-3}}$ on average within a distance of $15\,\mathrm{m}$ at STA, SCM and ROS respectively.

At all sites, traffic emissions are the biggest contributor to air pollution. Since the variability of traffic emissions does not change significantly from one day to the other (apart from the weekends), most of the concentration variability is attributable to changes in weather conditions, and this variability is generally well captured by the simulations. For instance, four events with strong winds (see wind speed in the upper panel of Fig. 4) and correspondingly low concentrations are well re-

produced. Nevertheless, inconsistencies in the selection of weather situations occasionally appear as suggested by sporadic peaks mostly at the site DUE, which appear in the simulations but not in the observations. Such events are most of the time not realistic and could point at a too frequent selection of very stable situations by the match-to-observation procedure. Such peaks may be amplified through the catalogue-based procedure, since particles emitted during a $1\,\mathrm{h}$ time span are fully

attributed to the same hour in the steady-state assumptions, while some particles are transported longer than $1\,\mathrm{h}$. Still, a few short peaks are observed and reproduced, demonstrating the ability of the system to capture short-term changes in weather conditions.

At the site HBR, $200\,\mathrm{m}$ above the altitude of the city centre, the concentrations are dominated by the background and are in most cases well reproduced. However, in some very specific situations,

the modelling system appears to miss some transport of pollution from the valley to higher altitudes. For instance, a pollution event was observed at all sites on Oct. 4[th] in Fig. 4. It was well reproduced by the model at all sites with the exception of HBR, where the model only marginally deviated from the background, suggesting underestimated vertical transport from the city centre, or a mismatch in simulated wind directions at higher altitudes.

**4.3.2 Complete two-year period**

The entire period of simulation, covering the years 2013 and 2014, is presented in Fig. 5 similar to Fig. 4 but as daily instead of hourly averages. Synoptic variability characterized by periods of strong air pollutant accumulation alternating with cleaner periods is superimposed on a seasonal cycle with systematically higher concentrations during winter. The synoptic variability, which is manifested by

varying contributions from emissions within the city with changing weather conditions, is very well reproduced with good FAC2 scores and high correlation coefficients for daily averages ($r = 0.86$ on average; see Tab. 2). At all sites, FAC2 and $r$ scores are 10–20% higher at the daily scale than at the hourly scale, indicating that the synoptic variability is better captured than the diurnal cycle. Events





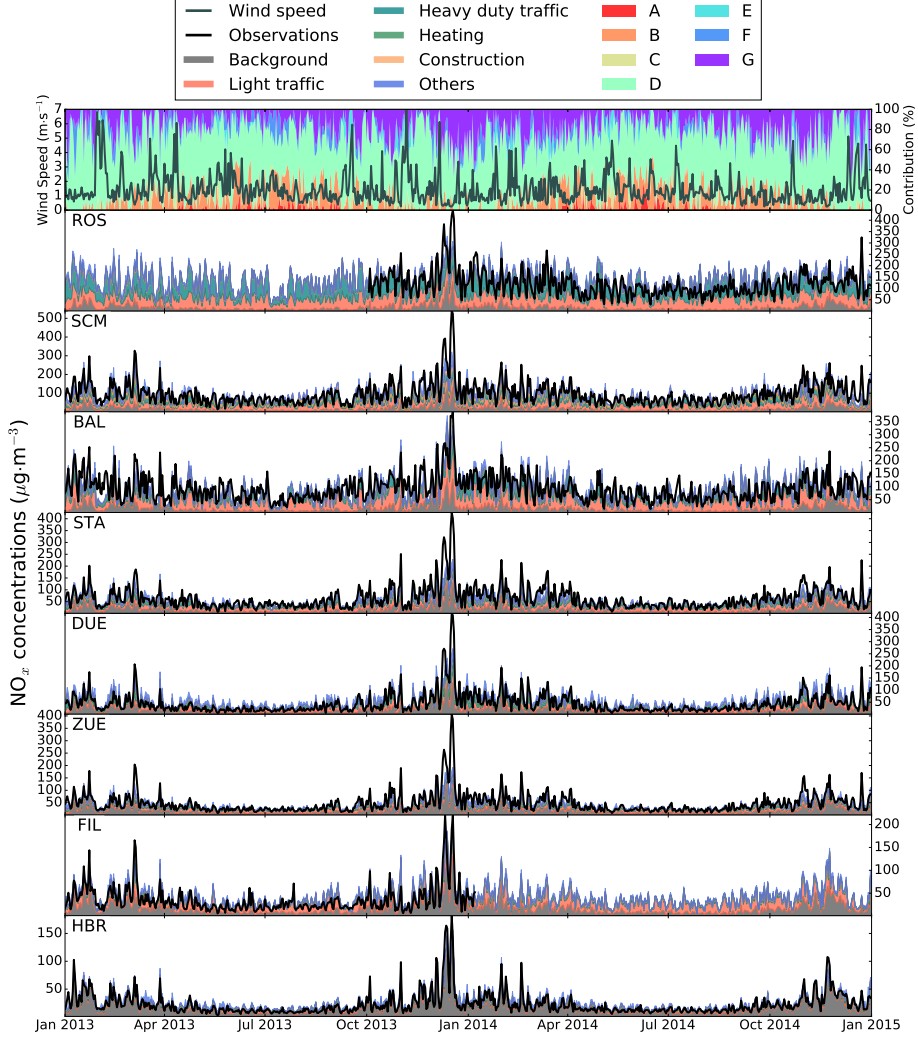

**Figure 5.** Daily observed and simulated concentrations at all observation sites for the years 2013 and 2014. Simulations are separated by contributions as in Fig. 4. (top) Daily averages of stability class contributions (A–G) and wind categories. Measurements at FIL were discontinued in January 2014, whereas the site ROS started being operated in October 2013.





of low concentrations are generally correlated with higher wind speeds whereas concentration peaks
are associated with particularly stable situations (stability classes E to G).

Although heating systems dominantly emit during winter time, they contribute only marginally
to the seasonal cycle of concentrations as simulated by the system. Indeed, all the observation sites
are located close to the ground at 3–4 m, while heating emissions occur on building roofs and fur-
ther rise after release, thus impacting ground concentrations only little. The seasonal cycle of $NO_x$
concentrations is, thus, mostly driven by the vertical stability of the atmosphere rather than by the
seasonal cycle of emissions. Convective situations are more frequent and pollutants are therefore
mixed more effectively during summer, reducing ground concentrations, whereas they accumulate
close to the surface in winter. Because of increased stable conditions, $NO_x$ pollution events tend to
be more severe in winter at all sites.

In December 2013, a particularly strong pollution event occurred. During this month, observed
daily averages reached up to $700\,\mu\mathrm{g}\cdot\mathrm{m}^{-3}$, with values above $200\,\mu\mathrm{g}\cdot\mathrm{m}^{-3}$ at all sites. These very
high concentrations are explained by the combination of regional recirculation of pollution (aver-
age background concentrations were $48\,\mu\mathrm{g}\cdot\mathrm{m}^{-3}$ for this month), low winds and highly stable con-
ditions enhancing the urban contribution. The modelling system reasonably fills the gap between
observed peaks and the background. However, it is unable to reach the observed peak concentrations
at most sites. The accumulation of pollutants in the city likely occurred over a time scale of several
hours, which our approach with steady-state hourly simulations is unable to reproduce. This ap-
proach underestimates air pollutant accumulation within the city during periods of particularly low
wind speeds, since this local accumulation cannot be represented by the rural background and as par-
ticles crossing the domain borders cannot re-enter the limited-area domain. General improvements
in the model set-up (e.g., higher spatial and temporal resolution and bigger domain) as well as in the
method (temporal hysteresis from one hour to the next ones) will be needed to better reproduce such
severe pollution events.

Figure 6 presents a general temporal evaluation (RMSEs and correlation coefficients) of the model
at all passive samplers. Biweekly $NO_2$ observations are compared to the corresponding $NO_2$ values
from the model obtained using the $NO_x$ to $NO_2$ conversion formula of Düring et al. (2011) (black
line in Fig. 3). For reference, correlation coefficients and RMSEs have been calculated as well for
different concentration intervals based on biweekly averages from continuous measurements of $NO_2$
and $NO_x$, represented by the blue and green line in Fig. 6, respectively. These lines, which are only
based on observations, provide a measure of the uncertainties implied by the $NO_x$–$NO_2$ conversion
alone and therefore of the best score achievable by the model. It appears that the two-weekly corre-
lation coefficients are very similar to the uncertainties implied by the conversion, suggesting a very
good agreement between observations and simulations, in agreement with the high correlations for
the daily averages at the continuous sites in Tab. 2. The RMSEs are on average two times larger
($3\,\mu\mathrm{g}\cdot\mathrm{m}^{-3}$) than the conversion uncertainties. Contrary to RMSEs computed from continuous sites,





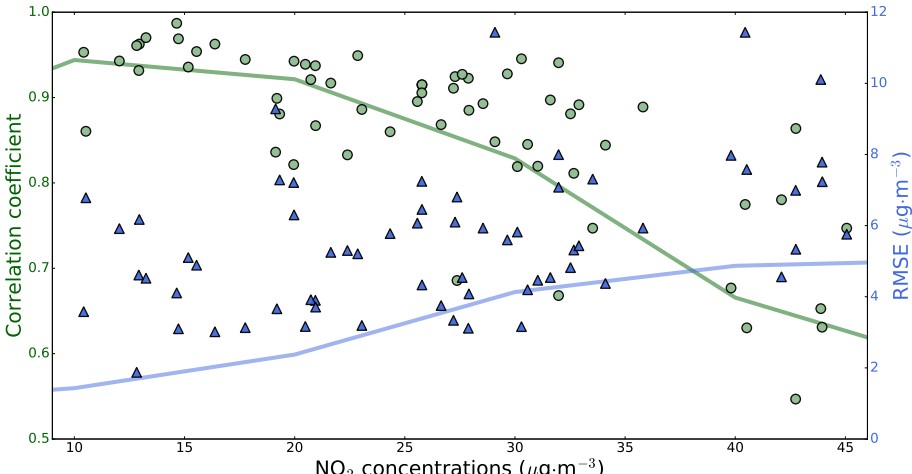

**Figure 6.** Correlation coefficients (green dots) and RMSE (blue triangles) at all passive samplers as a function of the mean $NO_2$ concentration at each site for the two-year period 2013–2014. $NO_x$ simulations are converted to $NO_2$ using the parametrized conversion of Düring et al. (2011). The green and blue line corresponds to the correlation coefficient and RMSE respectively, as computed for 5 $\mu g \cdot m^{-3}$ wide sliding concentration intervals by comparing two-week averages of continuous $NO_2$ observations with converted equivalents of observed $NO_x$.

RMSEs from converted simulations at passive sites do not show any clear dependency on concentration levels. This may partly be attributable to the systematic bias at the background site TAE for low concentrations (see Sect. 4.2).

### 4.3.3 Diurnal cycle of concentrations

Beyond the seasonal and daily variability, the diurnal cycle of concentrations plays a key role in assessing the exposure of the population to air pollution as people commute and spend their day at different locations within the city. The mean diurnal cycles of observations and simulations are presented in Fig. 7. Here, only weekdays are discussed as the diurnal cycle of emissions is more pronounced than during weekends. Consistent with the observations, the simulated concentrations 515 are higher at daytime than during night, with a morning peak at all sites and a late afternoon peak at some sites.

The morning rush hour leads to a stronger peak as the atmosphere is usually more stably stratified at this time of the day than during the evening rush hour. At most sites, both in the model and the observations, the peak occurs at 7 a.m. UTC, which corresponds to 8 a.m local time in winter 520 and 9 a.m. local time in summer. At HBR, the simulated and observed peaks happen later than at other sites. However, the observed peak is delayed by about one hour more than the simulated one. As mentioned in Sect. 4.3.1, HBR is an elevated site with no significant emissions nearby. Thus,





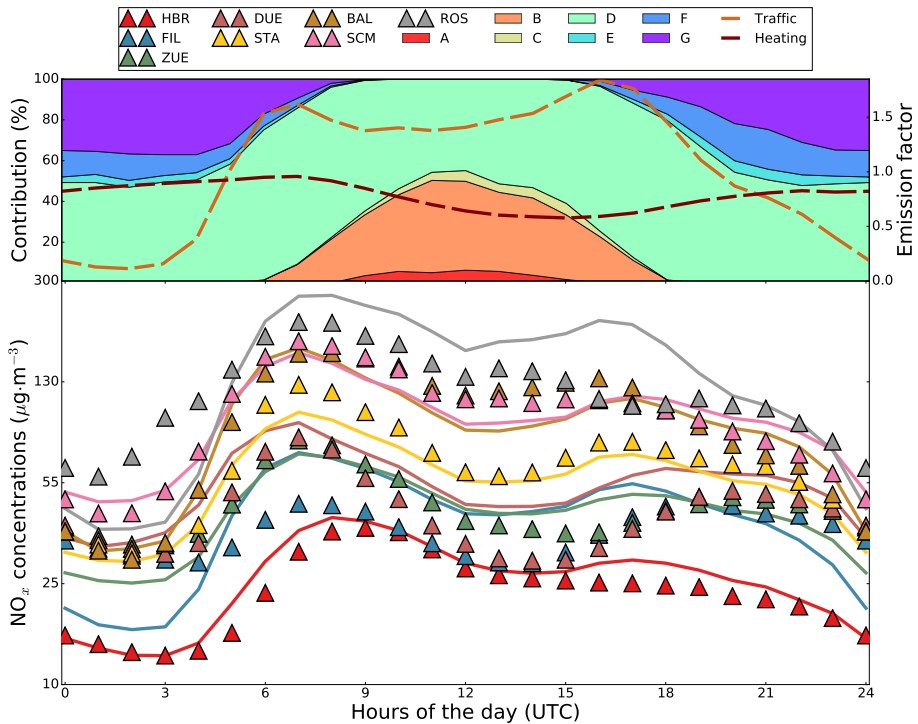

**Figure 7.** Two-year mean diurnal cycle of concentrations, traffic emissions and meteorology during weekdays.
(top) Contributions of stability classes (A very unstable, D neutral, G very stable) to the hourly meteorology.
The dashed lines are the emission profiles for light traffic and heating emissions. (bottom) Diurnal cycle of
observed (triangles) and simulated $NO_x$ concentrations (solid lines) at continuous measurement sites.

pollution emitted in the morning in the city appears to be transported to the site with a delay of 2–3
hours, while in the model, the steady-state assumption makes pollutants to be transported virtually

faster.

After the morning peak, the observed concentrations follow three possible paths: a steady decrease
until the night time minimum (ROS, SCM, HBR), an afternoon plateau with a small peak around
4 p.m. (STA, BAL) at the same time as the afternoon traffic peak, or a plateau with a late evening
concentration peak around 8–9 p.m. (FIL, DUE, ZUE). These patterns are correctly reproduced by

the model at the sites HBR, BAL, STA and DUE. At the other sites, the model simulates an after-
noon plateau followed by a small peak from traffic contributions not consistent with observations.
The uniform scaling of traffic emissions is a strong simplification that likely contributes to the dis-
crepancies between simulated and observed diurnal profiles at some sites: a closer analysis of the
data from the 89 traffic counters, for example, showed that the traffic intensity remains constantly

high at daytime in the city center whereas there are clear morning and evening peaks in the outer dis-



tricts. In addition, some streets are more intensively used by incoming traffic (strong morning peak), others by outgoing traffic (strong evening peak). The late evening peak at 8–9 p.m. UTC. could be explained by late emissions such as a surge in domestic heating before the night not accounted for in our system. A second explanation for the absence of a late-evening peak in the simulations could be
twofold: first, $NO_x$ is transported in the model as a passive tracer whereas in reality it is depleted by reaction with OH radicals. OH concentrations are highest when $NO_x$ is relatively low and solar radiation large (Ren et al., 2003), i.e., on sunny summer afternoons. The $NO_x$ lifetime is then reduced to about 2–4 hours (e.g., Liu et al., 2016), and not accounting for this depletion will contribute to a positive model bias in the afternoon as seen especially at sites in the lower range of the concentration
levels. Second, the observed late evening peak occurs well after the evening rush hour, suggesting that it is an integrated response due to accumulation of $NO_x$ over several hours. Such effects are not represented by our steady-state approach where the concentrations are solely determined by the emissions of the actual hour.

    During the night, observed concentrations at the sites BAL, DUE, ZUE, FIL and STA converge to
a similar level, which is well reproduced by the model at BAL, DUE, and STA, but underestimated at ZUE and especially at FIL. A systematic underestimation at night is also observed at ROS, especially in the early morning hours. The sites ROS and FIL are located next to important traffic corridors which might be used during the night more heavily than other roads including heavy duty traffic. Heavy duty traffic is not allowed in Switzerland at night between 22.00 and 05.00 local time (20.00
– 03.00 UTC in summer) but uses the early morning before the main rush hour intensively. Therefore, heavy duty traffic emissions close to these sites might be underestimated in our system. A second explanation can also be the missing accumulation of pollutants at night similar to the late evening peak. Under stable nocturnal situations, pollutants are slowly dispersed and remain longer in the domain of simulation than during the day. Accounting for air pollution accumulation over more than
one hour would help increase the nocturnal low concentrations.

## 5    Discussion and conclusions

A catalogue-based approach computed with the nested simulation system GRAMM/GRAL was applied to simulating hourly $NO_x$ concentration maps at 10 m resolution for the years 2013–2014. The modelling system was evaluated with 8 continuous $NO_x$ measurements sites and 65 $NO_2$ passive
sampler sites. The overall model performance was compliant with the objective criteria of the European air quality modelling framework FAIRMODE at both the hourly and the daily scale for most stations. The temporal variability of concentrations was well reproduced at the hourly, daily, two-weekly and seasonal scales. This can be explained to a minor extent by the proper representation of the variability in emissions, especially at the diurnal time scale, but it is mostly due to the successful
representation of the meteorological variability by the catalogue-based approach. The diurnal cycle





of concentrations, which is particularly critical to reproduce the pollution exposure of individuals commuting in the city, is largely consistent with the observations despite a systematic underestimation of concentrations during the night at some locations. The generally good performance of our system, on top of the reasonable computation costs and its flexibility (making it possible to carry out numerous simulation scenarios as it was done in Berchet et al., 2017), makes it a very suitable solution for designing informed air quality plans urgently needed at the city scale (Miranda et al., 2015).

Recent progress in parametrized approaches allows standard urban air pollution models to reach performances at the yearly or even monthly scale (e.g., ADMS-urban system; Dedele and Miskinyte, 2015) comparable to our approach. The model accuracy at diurnal to daily time scales, however, have hardly ever been analysed, which makes it very difficult to place our results in context but also demonstrates the uniqueness of the simulations presented in this work. At shorter temporal scales, our modelling system is still out-performed by very high resolution CFD models (Kumar et al., 2015a), but these systems are limited to small domains and short periods of time. Less complex systems such as SIRANE (Soulhac et al., 2012), solving the high-resolution flow only in street canyons and approximating the dispersion above the urban canopy as Gaussian plumes, perform similarly to our model, albeit at higher computational costs, limiting their application to periods of typically a few weeks only. Despite the usage of an extremely detailed emission inventory, our simulations are still significantly limited by the representation of emissions, since only standard temporal profiles were applied to most sources, which are unable to capture the large temporal dynamics of real emissions. Real-time emission models, informed for instance by mobile phone data and dense sensor networks could further advance the representation of emissions variability in the future. Such improved inputs proved to significantly increase the performance in other models (e.g., Soulhac et al., 2012; Borrego et al., 2016). The main gain of the catalogue-based method is, above all, the reduced computational cost allowing for high-resolution simulations of long time periods with a time resolution down to one hour. Further developments are required to improve this approach by replacing fixed traffic-emission patterns by transient ones, obtained with suitable traffic models.

A general overestimation of concentrations was found at all sites in our model, mostly related to insufficient dispersion in the model as well as to unrealistic accumulation of pollutants near buildings façades, which have a strong impact on simulations with the chosen 10 m horizontal resolution. The apparently too low dispersion may be related to the fact that traffic-induced turbulence is only crudely represented. Some limitations of the catalogue-based method were revealed, which are attributable to the steady-state assumption and the limited model domain. Particles that have been transported in the city for more than an hour are assigned to the same hour they were released in the current version of the system. Future versions should account for the particle transport age, which can be made accessible in the GRAL model outputs. This would likely smooth out some of the unrealistic short peaks produced in our simulations. A long residence time of particles in the simulation domain



can also have implications in terms of chemistry. $NO_x$ oxidation by OH radicals was neglected in our system, as the typical lifetime of $NO_x$ in the atmosphere is never shorter than a few hours (during

sunny summer afternoon; Liu et al., 2016). Accounting for long residence times in the simulation domain may allow us to compute simplified chemical reactions within the frame of the catalogue-based approach.

As demonstrated in this study, our model system produces a very realistic representation of the spatial distribution and temporal variability of $NO_x$ in the city, which makes it a highly suitable tool

for policy makers. The city of Zurich is indeed implementing the system as a new tool for improved air pollution control and urban planning. So far, our simulations have been generated without any input from actual air pollution measurements. Incorporating such observations through data assimilation and machine learning methods could further enhance the quality of the model predictions and even better satisfy the requirements of epidemiological studies, which need to be based on ac-

curate, unbiased data. The selection of weather situations in the catalogue could also benefit from assimilating concentration observations in the system, instead of meteorological observations only. A fully integrated high-resolution modelling system would enable short- and long-term pollutant and greenhouse gas monitoring in cities for subsequent use in the development of mitigation strategies.

**Code and data availability**

The system GRAMM/GRAL is made available by the Technische Universität Graz on the following webpage: lampx.tugraz.at/ gral/index.php. The catalogue-based method is fully described in Berchet et al. (2017) and related Python scripts can be requested to the corresponding author.

*Acknowledgements.*  We thank the City of Zurich for the building and emission inventory in Zurich, as well as their pollution monitoring network. We thank the cantonal environment office for sharing emission information

and pollution observations We thank MeteoSwiss and the Swiss Ministry for the Environment for providing data from their permanent measurement networks. This work was financed by the Swiss National Fund in the framework of the NanoTera project OpenSense2 and by the City of Zurich.



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
