# Peer review of "Evaluation of high-resolution GRAMM/GRAL NOx simulations over the city of Zurich, Switzerland"

_Geoscientific Model Development, 2017_

## Short Comment (SC1) · 8 Jun 2017

Dear authors,

in my role as Executive editor of GMD, I would like to bring to your attention our Editorial version 1.1:

http://www.geosci-model-dev.net/8/3487/2015/gmd-8-3487-2015.html

This highlights some requirements of papers published in GMD, which is also available on the GMD website in the 'Manuscript Types' section:

http://www.geoscientific-model-development.net/submission/manuscript_types.html

[Figure]

In particular, please note that for your paper, the following requirement has not been met in the Discussions paper:

- "The main paper must give the model name and version number (or other unique identifier) in the title."

Therefore, please add the version numbers of GRAMM/GRAL in the title of your article in your revised submission to GMD.

Additionally, I'd like to thank you for making the code publicly available. However, websites are often subject to change. Therefore the executive Editors of GMD highly recommend to make the exact code version to which the paper referes available via an archiving system providing a DOI (e.g. Zenodo). In this way, the code version and the paper are perfectly linked to each other.

Yours,

Astrid Kerkweg

<hr>

---

## Referee Comment (RC1) · Anonymous Referee #1 · 14 Jun 2017

**1. Overall quality of the discussion paper:**

The quality of the scientific content of the paper is very good. The authors describe the methodologies and data sets they use. The structure is good and makes the paper easy to read. Analysis is conducted with different statistical tools and the data is analyzed with respect to temporal and spatial properties with comprehensive measurement data sets. The authors also acknowledge that the city of Zürich is using their model for air pollution control, which adds value to the scientific contents.

**2. Individual scientific questions/issues:**

Line 13: “crude presentation of traffic induced turbulence” - is this assumption? As it is

discovered at many stations at least other reasons should be mentioned here.

GMDD

Line 93: I do not think the terminology “background winds” is right at this point, should probably be replaced by “large scale flow”.

Line 120: Does the modeling setup consider a memory of the model with respect to temporal evolution? Are pollutants e.g. transported according to wind turns at consecutive time steps?

Line 125: Are the background values assumed constant values for the whole domain?

Line 206: It is not explained how the modeling system is obtaining the meteorological initial- and boundary conditions? Is there an interface to a larger scale- or global model?

Line 357: In the paragraph starting at this line, the authors describe that the model overestimates the observations, which is furthermore omitted by just taking the minimum concentration values within a certain distance from the receptor. As the comparison of concentrations in a model grid cell with point-measurements causes deviations by nature this method can be applied in order to have more realistic results. However I would recommend to describe this procedure already in the beginning of the chapter and also exchange the values in Table 2 by those ones. This is not a requirement for publication from my side but an advise to improve the paper.

**3. Technical corrections: -**

---

## Referee Comment (RC2) · M. Schaap (Referee) · 21 Jul 2017

Dear authors,

I would like to congratelate you with the high quality paper submitted to GMD. It was a pleasure to read the paper. The modelling strategy and resulst are logically ordered and the red line can be followed very well. The paper deserves to be published with a few minor changes.

The few point I would like to raise are:

- line 476: The statements concern the local contribution. Could it be different for the

regional background?

-Line 608: Concerning the potential chemistry the night time N2O5 route could be relevant at medium range NOx levels as well. This reaction may also limit the life times to a few hours.

- Could you add a few lines on the possibility to assess the NO2 levels based on your calculations and using ozone data? Would the assessment of NO2 complicate the steady state classification scheme?

-Concerning emission modelling other emission categories also include temperature effect that my impact local contributions such as temperature and cold starts, etc.

- In a follow-up study I would to be curious in how far the system is able to reflect concentration variability with meteorological parameters that are not usd inthe classification system directly, such as ambient temperature.

again it was a pleasure to learn about your very interesting model approach! Martijn Schaap
* * *

---

## Author Comment (AC1) · 7 Aug 2017

We thank the reviewers for their useful comments and present in the attached document our replies and an updated manuscript with minor revisions highlighted in the text.

Please also note the supplement to this comment:
https://www.geosci-model-dev-discuss.net/gmd-2017-102/gmd-2017-102-AC1-supplement.pdf

---

## Editor Comment (EC1) · T. Butler (Editor) · 8 Aug 2017

Dear Authors,

Many thanks for your detailed response to the reviewer comments on your paper. Unfortunately I do not see any response to the short comment from the GMD executive editor Astrid Kerkweg.

Before considering your revised manuscript for publication in GMD, I would like to ask you to respond to the short comment. In particular, please consider making your model code available from a permanent archive, or discuss briefly why this has not been done.

[Figure]

Best regards, Tim Butler

---

## Author Comment (AC2) · 9 Aug 2017

We acknowledge the comments on editorial requirements by GMD.

The title of the final version of our manuscript is modified to: "Evaluation of high-resolution GRAMM/GRAL (v15.12/v14.8) NO$_X$ simulations over the city of Zurich, Switzerland"

In addition, as suggested, we made our Python library publicly available through the service Zenodo and reference to the associated doi in the Section "Code and Data availability".

---

## Author Response (AR2)

We would like to thank the reviewers for their useful comments on our manuscript. We suggest below minor revisions to improve our manuscript following their recommendations.

In the following, the comments from reviewer 1 are in bold red; those from Martijn Schaap (reviewer 2) in bold green; our responses are in normal font.

**Comments by referee 1:**

- **Line 13: "crude presentation of traffic induced turbulence" - is this assumption? As it is discovered at many stations at least other reasons should be mentioned here.**
  Other explanations are indeed possible, such as too low dispersion near buildings.
  We add such other likely explanations in the abstract, l. 14.
  We expanded the sentence to "possibly due to a crude representation of traffic-induced turbulence and to underestimated dispersion in the vicinity of buildings"

- **Line 93: I do not think the terminology "background winds" is right at his point, should probably be replaced by "large scale flow".**
  We agree with the reviewer. "background winds" is replaced by "large-scale flow", l.94.

- **Line 120: Does the modeling setup consider a memory of the model with respect to temporal evolution? Are pollutants e.g. transported according to wind turns at consecutive time steps?**
  Hourly simulations are based on reference steady-state computations. There is thus no memory from previous time steps and wind turns are not accounted for at the moment. We are aware that this poses a limitation in some cases and, therefore, future work should attempt to incorporate a memory component by prolonging the virtual particles of the previous time step and making use of the age of the virtual particles, which is accessible in this Lagrangian setup.
  This limitation will be further detailed in l.126-132 as follows:
  "Generating concentration fields directly from matched weather situations and steady-state dispersion simulations implicitly attributes the influence of emissions at a specific time stamp entirely to the same hour, even though some virtual particles remain longer than one hour in the domain. It thus prevents accounting for the accumulation of air pollutants over subsequent hours and for flow changes during such periods of accumulation, but performs well in most cases as demonstrated in Sect. 4."

- **Line 125: Are the background values assumed constant values for the whole domain?**
  The background values are indeed assumed constant over the entire domain. Even though this approach seems valid regarding the generally good results of our study, the background might be different from one valley to another, especially under stable conditions. We are currently working on a refinement of the background treatment in collaboration with experts from the city of Zurich with the aim to account for both vertical and horizontal gradients by incorporating more background observation sites.
  This will be explained by the following additional sentence, l. 134-137:
  "This approach implicitly assumes that background concentrations are uniform over the whole modelling domain, even though small horizontal and vertical gradients may exist, especially in our set-up with complex topography. A future refinement of the approach would thus be desirable."

- **Line 206: It is not explained how the modeling system is obtaining the meteorological initial- and boundary conditions? Is there an interface to a larger scale- or global model?**
No, the initial and boundary conditions are not obtained from a large-scale model but are idealized profiles that depend on the selected meteorological situation: For each stability class and largescale wind speed and direction, a vertical profile of wind and temperature is generated and used as initial condition and, during the GRAMM simulation, as boundary condition. The complete procedure is detailed in Berchet et al. (2016) and Oettl (2015, 2016). We now refer explicitly to these publications in Sect. 2.2, l.167-168

- **Line 357: In the paragraph starting at this line, the authors describe that the model overestimates the observations, which is furthermore omitted by just taking the minimum concentration values within a certain distance from the receptor. As the comparison of concentrations in a model grid cell with point-measurements causes deviations by nature this method can be applied in order to have more realistic results. However I would recommend to describe this procedure already in the beginning of the chapter and also exchange the values in Table 2 by those ones. This is not a requirement for publication from my side but an advice to improve the paper.**
The current version of Table 2 shows both versions of the data, and we think it is important to keep it this way to maintain full transparency. However, we agree that we should better prepare for this and therefore expanded the end of Sect. 2.3 as follows: "Scores are presented for two different simulated time series representing the concentrations at the exact location of the observation sites (reference), and the minimum of all concentrations in a radius of 15m horizontally and 2m vertically (minimum), respectively. The motivation for this will be presented in Sect. 4.1."

**Comments by referee 2,** Martijn Schaap:

- **Line 476: The statements concern the local contribution. Could it be different for the regional background?**
Similar arguments hold for the background, but in addition, the longer lifetime of NOx will be another factor contributing to an elevated background in this winter. This is mentioned more clearly in l.489-493:
The background concentrations partly explain the seasonal cycle of city concentrations, especially in the background site HBR and during the strong pollution event in December 2013. Local contributions from traffic emissions dominate the seasonal variability at all other sites even though traffic emissions do not change significantly from one month to another.

- **Line 608: Concerning the potential chemistry the night time $N_2O_5$ route could be relevant at medium range $NO_x$ levels as well. This reaction may also limit the life times to a few hours.**
We thank the reviewer for pointing at this aspect. We changed the sentence l.632-633

to "NO$_x$ depletion due to oxidation by OH radicals or night time N$_2$O$_5$ chemistry was neglected ..."

- **Could you add a few lines on the possibility to assess the NO$_2$ levels based on your calculations and using ozone data? Would the assessment of NO$_2$ complicate the steady state classification scheme?**
  Simulating NO$_2$ concentrations from NO$_x$ simulations is a very challenging problem. The steady-state approach should not limit such task as NO$_2$-O$_3$ chemistry is very fast. Simulating chemistry was attempted by accounting for the age of virtual particles in our simulations. The biggest challenge seems to be the assessment of O3 values to constrain chemical reactions. Oettl and Uhrner (2011) proposed an update to the GRAL system simulating explicitly 3D ozone fields and using them for converting NOx to NO2, but such approach loses the advantage of the catalogue method and is very costly.
  Possible approach to simulated NO$_2$ concentrations from NO$_x$ simulations is mentioned in l.652-659

- **Concerning emission modelling other emission categories also include temperature effect that may impact local contributions such as temperature and cold starts, etc.**
  We fully agree and modified the sentence on "Real-time emission models" l. 605-612 as follows:
  "Real-time emission models accounting for the influence of actual activities such as traffic density and energy consumption or environmental factors such as outdoor temperature affecting not only heating but also cold-start traffic emissions (as suggested by the Handbook of Emission Factors for Road Transport v3.3; Keller et al., 2017) could further advance the representation of emissions variability in the future. Such emission models and complementary influence on emission variability could be informed for instance by mobile phone data and sensor networks."

- **In a follow-up study I would to be curious in how far the system is able to reflect concentration variability with meteorological parameters that are not used in the classification system directly, such as ambient temperature.**
  Outdoor temperatures are already accounted for to modulate heating emissions through heating-degree days. Using such environment based variability of heating emissions proved to be better than using a generic seasonal cycle. Additional information would also be useful as discussed in reply above.

**Editorial comments:**

We acknowledge the comments on editorial requirements by GMD. The title of the final version of our manuscript is modified to: "Evaluation of high-resolution GRAMM/GRAL (v15.12/v14.8) NOx simulations over the city of Zurich, Switzerland"
In addition, as suggested, we made our Python library publicly available through the service Zenodo and reference to the associated doi in the Section "Code and Data availability".

**Additional revisions:**

Following internal discussions, the following small revisions were implemented in the figures to improve their quality:

- Figure 2: the line thickness of the city borders has been increased
- Figure 3: moved legend and enlarged circles for continuous observations in the right-hand figure
- Figure 4: "Traffic" replaced by "Traffic factor" in the legend to make clear that it refers to emission factor and not concentrations
- Figure 7: The legend is re-ordered to be more compact

[revised manuscript text omitted]